# Optimal spectral transportation with application to music transcription

**Rémi Flamary**
Université Côte d'Azur, CNRS, OCA
`remi.flamary@unice.fr`

**Cédric Févotte**
CNRS, IRIT, Toulouse
`cedric.fevotte@irit.fr`

**Nicolas Courty**
Université de Bretagne Sud, CNRS, IRISA
`courty@univ-ubs.fr`

**Valentin Emiya**
Aix-Marseille Université, CNRS, LIF
`valentin.emiya@lif.univ-mrs.fr`

## Abstract

Many spectral unmixing methods rely on the non-negative decomposition of spectral data onto a dictionary of spectral templates. In particular, state-of-the-art music transcription systems decompose the spectrogram of the input signal onto a dictionary of representative note spectra. The typical measures of fit used to quantify the adequacy of the decomposition compare the data and template entries frequency-wise. As such, small displacements of energy from a frequency bin to another as well as variations of timbre can disproportionally harm the fit. We address these issues by means of optimal transportation and propose a new measure of fit that treats the frequency distributions of energy holistically as opposed to frequency-wise. Building on the harmonic nature of sound, the new measure is invariant to shifts of energy to harmonically-related frequencies, as well as to small and local displacements of energy. Equipped with this new measure of fit, the dictionary of note templates can be considerably simplified to a set of Dirac vectors located at the target fundamental frequencies (musical pitch values). This in turns gives ground to a very fast and simple decomposition algorithm that achieves state-of-the-art performance on real musical data.

## 1    Context

Many of nowadays spectral unmixing techniques rely on non-negative matrix decompositions. This concerns for example hyperspectral remote sensing (with applications in Earth observation, astronomy, chemistry, etc.) or audio signal processing. The spectral sample $\mathbf{v}_n$ (the spectrum of light observed at a given pixel $n$, or the audio spectrum in a given time frame $n$) is decomposed onto a dictionary $\mathbf{W}$ of elementary spectral templates, characteristic of pure materials or sound objects, such that $\mathbf{v}_n \approx \mathbf{W}\mathbf{h}_n$. The composition of sample $n$ can be inferred from the non-negative expansion coefficients $\mathbf{h}_n$. This paradigm has led to state-of-the-art results for various tasks (recognition, classification, denoising, separation) in the aforementioned areas, and in particular in music transcription, the central application of this paper.

In state-of-the-art music transcription systems, the spectrogram $\mathbf{V}$ (with columns $\mathbf{v}_n$) of a musical signal is decomposed onto a dictionary of pure notes (in so-called multi-pitch estimation) or chords. $\mathbf{V}$ typically consists of (power-)magnitude values of a regular short-time Fourier transform (Smaragdis and Brown, 2003). It may also consists of an audio-specific spectral transform such as the Mel-frequency transform, like in (Vincent et al., 2010), or the Q-constant based transform, like in (Oudre et al., 2011). The success of the transcription system depends of course on the adequacy of the time-frequency transform & the dictionary to represent the data $\mathbf{V}$. In particular, the matrix $\mathbf{W}$ must

be able to accurately represent a diversity of real notes. It may be trained with individual notes using annotated data (Boulanger-Lewandowski et al., 2012), have a parametric form (Rigaud et al., 2013) or be learnt from the data itself using a harmonic subspace constraint (Vincent et al., 2010).

One important challenge of such methods lies in their ability to cope with the variability of real notes. A simplistic dictionary model will assume that one note characterised by fundamental frequency $\nu_0$ (e.g., $\nu_0 = 440$ Hz for note A$_4$) will be represented by a spectral template with non-zero coefficients placed at $\nu_0$ and at its multiples (the *harmonic frequencies*). In reality, many instruments, such as the piano, produce musical notes with either slight frequency misalignments (so-called *inharmonicities*) with respect to the theoretical values of the fundamental and harmonic frequencies, or amplitude variations at the harmonic frequencies with respect to recording conditions or played instrument (variations of *timbre*). Handling these variabilities by increasing the dictionary with more templates is typically unrealistic and adaptive dictionaries have been considered in (Vincent et al., 2010; Rigaud et al., 2013). In these papers, the spectral shape of the columns of $\mathbf{W}$ is adjusted to the data at hand, using specific time-invariant semi-parametric models. However, the note realisations may vary in time, something which is not handled by these approaches. This work presents a new spectral unmixing method based on optimal transportation (OT) that is fully flexible and remedies the latter difficulties. Note that Typke et al. (2004) have previously applied OT to notated music (e.g., score sheets) for search-by-query in databases while we address here music transcription from audio spectral data.

## 2 A relevant baseline: PLCA

Before presenting our contributions, we start by introducing the PLCA method of Smaragdis et al. (2006) which is heavily used in audio signal processing. It is based on the Probabilistic Latent Semantic Analysis (PLSA) of Hofmann (2001) (used in text retrieval) and is a particular form of non-negative matrix factorisation (NMF). Simplifying a bit, in PLCA the columns of $\mathbf{V}$ are normalised to sum to one. Each vector $\mathbf{v}_n$ is then treated as a discrete probability distribution of "frequency quanta" and is approximated as $\mathbf{V} \approx \mathbf{WH}$. The matrices $\mathbf{W}$ and $\mathbf{H}$ are of size $M \times K$ and $K \times N$, respectively, and their columns are constrained to sum to one. As a result, the columns of the approximate $\hat{\mathbf{V}} = \mathbf{WH}$ sum to one as well and each distribution vector $\mathbf{v}_n$ is as such approximated by the counterpart distribution $\hat{\mathbf{v}}_n$ in $\hat{\mathbf{V}}$. Under the assumption that $\mathbf{W}$ is known, the approximation is found by solving the optimisation problem defined by

$$\min_{\mathbf{H} \geq 0} D_{\mathrm{KL}}(\mathbf{V}|\mathbf{WH}) \quad \text{s.t} \quad \forall n, \|\mathbf{h}_n\|_1 = 1, \tag{1}$$

where $D_{\mathrm{KL}}(\mathbf{v}|\hat{\mathbf{v}}) = \sum_i v_i \log(v_i/\hat{v}_i)$ is the KL divergence between discrete distributions, and by extension $D_{\mathrm{KL}}(\mathbf{V}|\hat{\mathbf{V}}) = \sum_n D_{\mathrm{KL}}(\mathbf{v}_n|\hat{\mathbf{v}}_n)$.

An important characteristic of the KL divergence is its separability with respect to the entries of its arguments. It operates a frequency-wise comparison in the sense that, at every frame $n$, the spectral coefficient $v_{in}$ at frequency $i$ is compared to its counterpart $\hat{v}_{in}$, and the results of the comparisons are summed over $i$. In particular, a small displacement in the frequency support of one observation may disproportionally harm the divergence value. For example, if $\mathbf{v}_n$ is a pure note with fundamental frequency $\nu_0$, a small inharmonicity that shifts energy from $\nu_0$ to an adjacent frequency bin will unreasonably increase the divergence value, when $\mathbf{v}_n$ is compared with a purely harmonic spectral template with fundamental frequency $\nu_0$. As explained in Section 1 such local displacements of frequency energy are very common when dealing with real data. A measure of fit invariant to small perturbations of the frequency support would be desirable in such a setting, and this is precisely what OT can bring.

## 3 Elements of optimal transportation

Given a discrete probability distribution $\mathbf{v}$ (a non-negative real-valued column vector of dimension $M$ and summing to one) and a *target* distribution $\hat{\mathbf{v}}$ (with same properties), OT computes a *transportation matrix* $\mathbf{T}$ belonging to the set $\Theta \stackrel{\text{def}}{=} \{\mathbf{T} \in \mathbb{R}_+^{M \times M} | \forall i, j = 1, \ldots, N, \sum_{j=1}^M t_{ij} = v_i, \sum_{i=1}^M t_{ij} = \hat{v}_j\}$. $\mathbf{T}$ establishes a bi-partite graph connecting the two distributions. In simple words, an amount (or, in typical OT parlance, a "mass") of every coefficient of vector $\mathbf{v}$ is transported to an entry of $\hat{\mathbf{v}}$. The sum of transported amounts to the $j^{th}$ entry of $\hat{\mathbf{v}}$ must equal $\hat{v}_j$. The value of $t_{ij}$ is the amount

transported from the $i^{th}$ entry of $\mathbf{v}$ to the $j^{th}$ entry of $\hat{\mathbf{v}}$. In our particular setting, the vector $\mathbf{v}$ is a distribution of spectral energies $v_1, \ldots, v_M$ at sampling frequencies $f_1, \ldots, f_M$.

Without additional constraints, the problem of finding a non-negative matrix $\mathbf{T} \in \Theta$ has an infinite number of solutions. As such, OT takes into account the *cost* of transporting an amount from the $i^{th}$ entry of $\mathbf{v}$ to the $j^{th}$ entry of $\hat{\mathbf{v}}$, denoted $c_{ij}$ (a non-negative real-valued number). Endorsed with this cost function, OT involves solving the optimisation problem defined by

$$\min_{\mathbf{T}} J(\mathbf{T}|\mathbf{v}, \hat{\mathbf{v}}, \mathbf{C}) = \sum\nolimits_{ij} c_{ij} t_{ij} \quad \text{s.t} \quad \mathbf{T} \in \Theta, \tag{2}$$

where $\mathbf{C}$ is the non-negative square matrix of size $M$ with elements $c_{ij}$. Eq. (2) defines a convex linear program. The value of the function $J(\mathbf{T}|\mathbf{v}, \hat{\mathbf{v}}, \mathbf{C})$ at its minimum is denoted $D_{\mathbf{C}}(\mathbf{v}|\hat{\mathbf{v}})$. When $\mathbf{C}$ is a symmetric matrix such that $c_{ij} = \|f_i - f_j\|_p^p$, where we recall that $f_i$ and $f_j$ are the frequencies in Hertz indexed by $i$ and $j$, $D_{\mathbf{C}}(\mathbf{v}|\hat{\mathbf{v}})$ defines a metric (i.e., a symmetric divergence that satisfies the triangle inequality) coined Wasserstein distance or earth mover's distance (Rubner et al., 1998; Villani, 2009). In other cases, in particular when the matrix $\mathbf{C}$ is not even symmetric like in the next section, $D_{\mathbf{C}}(\mathbf{v}|\hat{\mathbf{v}})$ is not a metric in general, but is still a valid measure of fit. For generality, we will refer to it as the "OT divergence".

By construction, the OT divergence can explicitly embed a form of invariance to displacements of support, as defined by the transportation cost matrix $\mathbf{C}$. For example, in the spectral decomposition setting, the matrix with entries of the form $c_{ij} = (f_i - f_j)^2$ will increasingly penalise frequency displacements as the distance between frequency bins increases. This precisely remedies the limitation of the separable KL divergence presented in Section 2. As such, the next section addresses variants of spectral unmixing based on the Wasserstein distance.

## 4  Optimal spectral transportation (OST)

**Unmixing with OT.**  In light of the above discussion, a direct solution to the sensibility of PLCA to small frequency displacements consists in replacing the KL divergence with the OT divergence. This amounts to solving the optimisation problem given by

$$\min_{\mathbf{H} \geq 0} D_{\mathbf{C}}(\mathbf{V}|\mathbf{WH}) \quad \text{s.t} \quad \forall n, \|\mathbf{h}_n\|_1 = 1, \tag{3}$$

where $D_{\mathbf{C}}(\mathbf{V}|\hat{\mathbf{V}}) = \sum_n D_{\mathbf{C}}(\mathbf{v}_n|\hat{\mathbf{v}}_n)$, $\mathbf{W}$ is fixed and populated with pure note spectra and $\mathbf{C}$ penalises large displacements of frequency support. This approach is a particular case of NMF with the Wasserstein distance, which has been considered in a face recognition setting by Sandler and Lindenbaum (2011), with subsequent developments by Zen et al. (2014) and Rolet et al. (2016). This approach is relevant to our spectral unmixing scenario but as will be discussed in Section 5 is on the downside computationally intensive. It also requires the columns of $\mathbf{W}$ to be set to realistic note templates, which is still constraining. The next two sections describes a computationally more friendly approach which additionally removes the difficulty of choosing $\mathbf{W}$ appropriately.

**Harmonic-invariant transportation cost.**  In the approach above, the harmonic modelling is conveyed by the dictionary $\mathbf{W}$ (consisting of comb-like pure note spectra) and the invariance to small frequency displacements is introduced via the matrix $\mathbf{C}$. In this section we propose to model both harmonicity and local invariance through the transportation cost matrix $\mathbf{C}$. Loosely speaking, we want to define a class of equivalence between musical spectra, that takes into account their inherent harmonic nature. As such, we essentially impose that a harmonic frequency (i.e., a close multiple of its fundamental) can be considered equivalent to its fundamental, the only target of multi-pitch estimation. As such, we assume that a mass at one frequency can be transported to a divisor frequency with no cost. In other words, a mass at frequency $f_i$ can be transported with no cost to $f_i/2$, $f_i/3$, $f_i/4$, and so on until sampling resolution. One possible cost matrix that embeds this property is

$$c_{ij} = \min_{q=1,\ldots,q_{\max}} (f_i - q f_j)^2 + \epsilon \delta_{q \neq 1}, \tag{4}$$

where $q_{\max}$ is the ceiling of $f_i/f_j$ and $\epsilon$ is a small value. The term $\epsilon \delta_{q \neq 1}$ favours the discrimination of octaves. Indeed, it penalises the transportation of a note of fundamental frequency $2\nu_0$ or $\nu_0/2$ to the spectral template with fundamental frequency $\nu_0$, which would be costless without this additive term. Let us denote by $\mathbf{C}_h$ the transportation cost matrix defined by Eq. (4). Fig. 1 compares $\mathbf{C}_h$

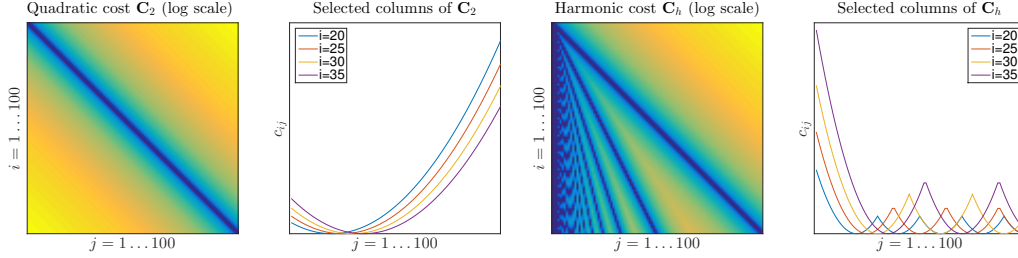

Figure 1: Comparison of transportation cost matrices $\mathbf{C}_2$ and $\mathbf{C}_h$ (full matrices and selected columns).

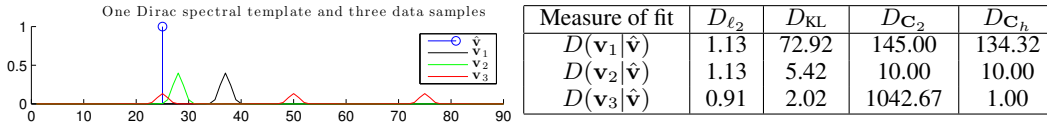

| Measure of fit | $D_{\ell_2}$ | $D_{\mathrm{KL}}$ | $D_{\mathbf{C}_2}$ | $D_{\mathbf{C}_h}$ |
|---|---|---|---|---|
| $D(\mathbf{v}_1\|\hat{\mathbf{v}})$ | 1.13 | 72.92 | 145.00 | 134.32 |
| $D(\mathbf{v}_2\|\hat{\mathbf{v}})$ | 1.13 | 5.42 | 10.00 | 10.00 |
| $D(\mathbf{v}_3\|\hat{\mathbf{v}})$ | 0.91 | 2.02 | 1042.67 | 1.00 |

Figure 2: Three example spectra $\mathbf{v}_n$ compared to a given template $\hat{\mathbf{v}}$ (left) and computed divergences (right). The template is a mere Dirac vector placed at a particular frequency $\nu_0$. $D_{\ell_2}$ denotes the standard quadratic error $\|\mathbf{x} - \mathbf{y}\|_2^2$. By construction of $D_{\mathbf{C}_h}$, sample $\mathbf{v}_3$ which is harmonically related to the template returns a very good fit with the latter OT divergence. Note that it does not make sense to compare output values of different divergences; only the relative comparison of output values of the same divergence for different input samples is meaningful.

to the more standard quadratic cost $\mathbf{C}_2$ defined by $c_{ij} = (f_i - f_j)^2$. With the quadratic cost, only local displacements are permissible. In contrast, the harmonic-invariant cost additionally permits larger displacements to divisor frequencies, improving robustness to variations of timbre besides to inharmonicities.

**Dictionary of Dirac vectors.** Having designed an OT divergence that encodes inherent properties of musical signals, we still need to choose a dictionary $\mathbf{W}$ that will encode the fundamental frequencies of the notes to identify. Typically, these will consist of the physical frequencies of the 12 notes of the chromatic scale (from note A to note G, including half-tones), over several octaves. As mentioned in Section 1, one possible strategy is to populate $\mathbf{W}$ with spectral note templates. However, as also discussed, the performance of the resulting unmixing method will be capped by the representativeness of the chosen set of templates.

A most welcome consequence of using the OT divergence built on the harmonic-insensitive cost matrix $\mathbf{C}_h$ is that we may use for $\mathbf{W}$ a mere set of Dirac vectors placed at the fundamental frequencies $\nu_1, \ldots, \nu_K$ of the notes to identify and separate. Indeed, under the proposed setting, a real note spectra (composed of one fundamental and multiple harmonic frequencies) can be transported with no cost to its fundamental. Similarly, a spectral sample composed of several notes can be transported to mixture of Dirac vectors placed at their fundamental frequencies. This simply eliminates the problem of choosing a representative dictionary! This very appealing property is illustrated in Fig. 2. Furthermore, the particularly simple structure of the dictionary leads to a very efficient unmixing algorithm, as explained in the next section. In the following, the unmixing method consisting of the combined use of the harmonic-invariant cost matrix $\mathbf{C}_h$ and of the dictionary of Dirac vectors will be coined "optimal spectral transportation" (OST).

At this level, we assume for simplicity that the set of $K$ fundamental frequencies $\{\nu_1, \ldots, \nu_K\}$ is contained in the set of sampled frequencies $\{f_1, \ldots, f_M\}$. This means that $\mathbf{w}_k$ (the $k^{th}$ column of $\mathbf{W}$) is zero everywhere except at some entry $i$ such that $f_i = \nu_k$ where $w_{ik} = 1$. This is typically not the case in practice, where the sampled frequencies are fixed by the sampling rate, of the form $f_i = 0.5(i/T)f_s$, and where the fundamental frequencies $\nu_k$ are fixed by music theory. Our approach can actually deal with such a discrepancy and this will be explained later in Section 5.

# 5 Optimisation

**OT unmixing with linear programming.** We start by describing optimisation for the state-of-the-art OT unmixing problem described by Eq. (3) and proposed by Sandler and Lindenbaum (2011). First, since the objective function is separable with respect to samples, the optimisation problem decouples with respect to the activation columns $\mathbf{h}_n$. Dropping the sample index $n$ and combining Eqs. (2) and (3), optimisation thus reduces to solving for every sample a problem of the form

$$\min_{\mathbf{h}\geq 0, \mathbf{T}\geq 0} \quad \langle \mathbf{T}, \mathbf{C} \rangle = \sum_{ij} t_{ij} c_{ij} \quad \text{s.t.} \quad \mathbf{T}\mathbf{1}_M = \mathbf{v}, \quad \mathbf{T}^\top \mathbf{1}_M = \mathbf{W}\mathbf{h}, \tag{5}$$

where $\mathbf{1}_M$ is a vector of dimension $M$ containing only ones and $\langle \cdot, \cdot \rangle$ is the Frobenius inner product. Vectorising the variables $\mathbf{T}$ and $\mathbf{h}$ into a single vector of dimension $M^2 + K$, problem (5) can be turned into a canonical linear program. Because of the large dimension of the variable (typically in the order of $10^5$), resolution can however be very demanding, as will be shown in experiments.

**Optimisation for OST.** We now assume that $\mathbf{W}$ is a set of Dirac vectors as explained at the end of Section 4. We also assume that $K < M$, which is the usual scenario. Indeed, $K$ is typically in the order of a few tens, while $M$ is in the order of a few hundreds. In such a setting $\hat{\mathbf{v}} = \mathbf{W}\mathbf{h}$ contains by design at most $K$ non-zero coefficients, located at the entries such that $f_i = \nu_k$. We denote this set of frequency indices by $\mathcal{S}$. Hence, for $j \notin \mathcal{S}$, we have $\hat{v}_j = 0$ and thus $\sum_i t_{ij} = 0$, by the second constraint of Eq. (5). Additionally, by the non-negativity of $\mathbf{T}$ this also implies that $\mathbf{T}$ has only $K$ non-zero columns, indexed by $j \in \mathcal{S}$. Denoting by $\widetilde{\mathbf{T}}$ this subset of columns, and by $\widetilde{\mathbf{C}}$ the corresponding subset of columns of $\mathbf{C}$, problem (5) reduces to

$$\min_{\mathbf{h}\geq 0, \widetilde{\mathbf{T}}\geq 0} \langle \widetilde{\mathbf{T}}, \widetilde{\mathbf{C}} \rangle \quad \text{s.t.} \quad \widetilde{\mathbf{T}}\mathbf{1}_K = \mathbf{v}, \quad \widetilde{\mathbf{T}}^\top \mathbf{1}_M = \mathbf{h}. \tag{6}$$

This is an optimisation problem of significantly reduced dimension $(M + 1)K$. Even more appealing, the problem has a simple closed-form solution. Indeed, the variable $\mathbf{h}$ has a virtual role in problem (6). It only appears in the second constraint, which *de facto* becomes a free constraint. Thus problem (6) can be solved with respect to $\widetilde{\mathbf{T}}$ regardless of $\mathbf{h}$, and $\mathbf{h}$ is then simply obtained by summing the columns of $\widetilde{\mathbf{T}}^\top$ at the solution. Now, the problem

$$\min_{\widetilde{\mathbf{T}}\geq 0} \langle \widetilde{\mathbf{T}}, \widetilde{\mathbf{C}} \rangle \quad \text{s.t.} \quad \widetilde{\mathbf{T}}\mathbf{1}_K = \mathbf{v} \tag{7}$$

decouples with respect to the rows $\underline{\tilde{t}}_i$ of $\widetilde{\mathbf{T}}$, and becomes, $\forall i = 1, \ldots, M$,

$$\min_{\underline{\tilde{t}}_i \geq 0} \sum_k \tilde{t}_{ik} \tilde{c}_{ik} \quad \text{s.t.} \quad \sum_k \tilde{t}_{ik} = v_i. \tag{8}$$

The solution is simply given by $\tilde{t}_{ik_i^\star} = v_i$ for $k_i^\star = \arg\min_k \{\tilde{c}_{ik}\}$, and $\tilde{t}_{ik} = 0$ for $k \neq k_i^\star$. Introducing the labelling matrix $\mathbf{L}$ which is everywhere zero except for indices $(i, k_i^\star)$ where it is equal to 1, the solution to OST is trivially given by $\hat{\mathbf{h}} = \mathbf{L}^\top \mathbf{v}$. Thus, under the specific assumption that $\mathbf{W}$ is a set of Dirac vectors, the challenging problem (5) has been reduced to an effortless assignment problem to solve for $\mathbf{T}$ and a simple sum to solve for $\mathbf{h}$. Note that the algorithm is independent of the particular structure of $\mathbf{C}$. In the end, the complexity per frame of OST reduces to $\mathcal{O}(M)$, which starkly contrasts with the complexity of PLCA, in the order $\mathcal{O}(KM)$ *per iteration*.

In Section 4, we assumed for simplicity that the set of fundamental frequencies $\{\nu_k\}_k$ was contained in the set of sampled frequencies $\{f_i\}_i$. As a matter of fact, this assumption can be trivially lifted in the proposed setting of OST. Indeed, we may construct the cost matrix $\widetilde{\mathbf{C}}$ (of dimensions $M \times K$) by replacing the target frequencies $f_j$ in Eq. (4) by the theoretical fundamental frequencies $\nu_k$. Namely, we may simply set the coefficients of $\widetilde{\mathbf{C}}$ to be $\tilde{c}_{ik} = \min_q (f_i - q\nu_k)^2 + \epsilon \delta_{q \neq 1}$, in the implementation. Then, the matrix $\widetilde{\mathbf{T}}$ indicates how each sample $\mathbf{v}$ is transported to the Dirac vectors placed at fundamental frequencies $\{\nu_k\}_k$, without the need for the actual Dirac vectors themselves, which elegantly solves the frequency sampling problem.

**OST with entropic regularisation ($\text{OST}_e$).** The procedure described above leads to a *winner-takes-all* transportation of all of $v_i$ to its cost-minimum target entry $k_i^\star$. We found it useful in

practice to relax this hard assignment and distribute energies more evenly by using the entropic regularisation of Cuturi (2013). It consists of penalising the fit $\langle \widetilde{\mathbf{T}}, \widetilde{\mathbf{C}} \rangle$ in Eq. (6) with an additional term $\Omega_e(\widetilde{\mathbf{T}}) = \sum_{ik} \tilde{t}_{ik} \log(\tilde{t}_{ik})$, weighted by the hyper-parameter $\lambda_e$. The negentropic term $\Omega_e(\widetilde{\mathbf{T}})$ promotes the transportation of $v_i$ to several entries, leading to a smoother estimate of $\widetilde{\mathbf{T}}$. As explained in the supplementary material, one can show that the negentropy-regularised problem is a Bregman projection (Benamou et al., 2015) and has again a closed-form solution $\hat{\mathbf{h}} = \mathbf{L}_e^\top \mathbf{v}$ where $\mathbf{L}_e$ is the $M \times K$ matrix with coefficients $l_{ik} = \exp(-\tilde{c}_{ik}/\lambda_e)/\sum_p \exp(-\tilde{c}_{ip}/\lambda_e)$. Limiting cases $\lambda_e = 0$ and $\lambda_e = \infty$ return the unregularised OST estimate and the maximum-entropy estimate $h_k = 1/K$, respectively. Because $\mathbf{L}_e$ becomes a full matrix, the complexity per frame of $\text{OST}_e$ becomes $\mathcal{O}(KM)$.

**OST with group regularisation ($\text{OST}_g$).** We have explained above that the transportation matrix $\mathbf{T}$ has a strong group structure in the sense that it contains by construction $M - K$ null columns, and that only the subset $\widetilde{\mathbf{T}}$ needs to be considered. Because a small number of the $K$ possible notes will be played at every time frame, the matrix $\widetilde{\mathbf{T}}$ will additionally have a significant number of null columns. This heavily suggests using group-sparse regularisation in the estimation of $\widetilde{\mathbf{T}}$. As such, we also consider problem (6) penalised by the additional term $\Omega_g(\widetilde{\mathbf{T}}) = \sum_k \sqrt{\|\widetilde{\mathbf{t}}_k\|_1}$ which promotes group-sparsity at column level (Huang et al., 2009). Unlike OST or $\text{OST}_e$, $\text{OST}_g$ does not offer a closed-form solution. Following Courty et al. (2014), a majorisation-minimisation procedure based on the local linearisation of $\Omega_g(\widetilde{\mathbf{T}})$ can be employed and the details are given in the supplementary material. The resulting algorithm consists in iteratively applying unregularised OST, as of Eq. (6), with the iteration-dependent transportation cost matrix $\widetilde{\mathbf{C}}^{(iter)} = \widetilde{\mathbf{C}} + \widehat{\mathbf{R}}^{(iter)}$, where $\widehat{\mathbf{R}}^{(iter)}$ is the $M \times K$ matrix with coefficients $\widetilde{r}_{ik}^{(iter)} = \frac{1}{2}\|\widetilde{\mathbf{t}}_k^{(iter)}\|_1^{-\frac{1}{2}}$. Note that the proposed group-regularisation of $\widetilde{\mathbf{T}}$ corresponds to a sparse regularisation of $\mathbf{h}$. This is because $h_k = \|\widetilde{\mathbf{t}}_k\|_1$ and thus, $\Omega_g(\widetilde{\mathbf{T}}) = \sum_k \sqrt{h_k}$. Finally, note that $\text{OST}_e$ and $\text{OST}_g$ can be implemented simultaneously, leading to $\text{OST}_{e+g}$, by considering the optimisation of the doubly-penalised objective function $\langle \widetilde{\mathbf{T}}, \widetilde{\mathbf{C}} \rangle + \lambda_e \, \Omega_e(\widetilde{\mathbf{T}}) + \lambda_g \, \Omega_g(\widetilde{\mathbf{T}})$, addressed in the supplementary material.

## 6 Experiments

**Toy experiments with simulated data.** In this section we illustrate the robustness, the flexibility and the efficiency of OST on two simulated examples. The top plots of Fig. 3 display a synthetic dictionary of 8 harmonic spectral templates, referred to as the "harmonic dictionary". They have been generated as Gaussian kernels placed at a fundamental frequency and its multiples, and using exponential dampening of the amplitudes. As everywhere in the paper, the spectral templates are normalised to sum to one. Note that the 8th template is the upper octave of the first one. We compare the unmixing performance of five methods in two different scenarios. The five methods are as follows. PLCA is the method described in Section 2, where the dictionary $\mathbf{W}$ is the harmonic dictionary. Convergence is stopped when the relative difference of the objective function between two iterations falls below $10^{-5}$ or the number of iterations (per frame) exceeds 1000. $\text{OT}_h$ is the unmixing method with the OT divergence, as in the first paragraph of Section 4, using the harmonic transportation cost matrix $\mathbf{C}_h$ and the harmonic dictionary. OST is like $\text{OT}_h$, but using a dictionary of Dirac vectors (placed at the 8 fundamental frequencies characterising the harmonic dictionary). $\text{OST}_e$, $\text{OST}_g$ and $\text{OST}_{e+g}$ are the regularised variants of OST, described at the end of Section 4. The iterative procedure in the group-regularised variants is run for 10 iterations (per frame).

In the first experimental scenario, reported in Fig. 3 (a), the data sample is generated by mixing the 1st and 4th elements of the harmonic dictionary, but introducing a small shift of the true fundamental frequencies (with the shift being propagated to the harmonic frequencies). This mimics the effect of possible inharmonicities or of an ill-tuned instrument. The middle plot of Fig. 3 (a), displays the generated sample, together with the "theoretical sample", i.e., without the frequencies shift. This shows how a slight shift of the fundamental frequencies can greatly impact the overall spectral distribution. The bottom plot displays the true activation vector and the estimates returned by the five methods. The table reports the value of the (arbitrary) error measure $\|\hat{\mathbf{h}} - \mathbf{h}_{\text{true}}\|_1$ together with the run time (on an average desktop PC using a MATLAB implementation) for every method. The results show that group-regularised variants of OST lead to best performance with very light computational

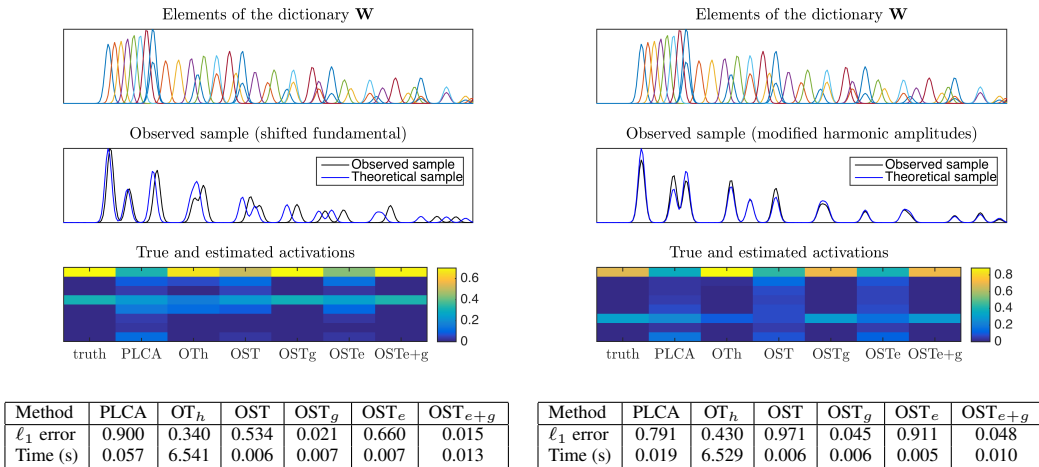

| (a) Unmixing with shifted fundamental frequencies | (b) Unmixing with wrong harmonic amplitudes |

Elements of the dictionary **W**

Observed sample (shifted fundamental)
— Observed sample
— Theoretical sample

True and estimated activations

truth  PLCA  OTh  OST  OSTg  OSTe  OSTe+g

| Method | PLCA | $OT_h$ | OST | $OST_g$ | $OST_e$ | $OST_{e+g}$ |
|---|---|---|---|---|---|---|
| $\ell_1$ error | 0.900 | 0.340 | 0.534 | 0.021 | 0.660 | 0.015 |
| Time (s) | 0.057 | 6.541 | 0.006 | 0.007 | 0.007 | 0.013 |

Elements of the dictionary **W**

Observed sample (modified harmonic amplitudes)
— Observed sample
— Theoretical sample

True and estimated activations

truth  PLCA  OTh  OST  OSTg  OSTe  OSTe+g

| Method | PLCA | $OT_h$ | OST | $OST_g$ | $OST_e$ | $OST_{e+g}$ |
|---|---|---|---|---|---|---|
| $\ell_1$ error | 0.791 | 0.430 | 0.971 | 0.045 | 0.911 | 0.048 |
| Time (s) | 0.019 | 6.529 | 0.006 | 0.006 | 0.005 | 0.010 |

Figure 3: Unmixing under model misspecification. See text for details.

burden, and without using the true harmonic dictionary. In the second experimental scenario, reported in Fig. 3 (b), the data sample is generated by mixing the 1st and 6th elements of the harmonic dictionary, with the right fundamental and harmonic frequencies, but where the spectral amplitudes at the latters do not follow the exponential dampening of the template dictionary (variation of timbre). Here again the group-regularised variants of OST outperforms the state-of-the-art approaches, both in accuracy and run time.

**Transcription of real musical data.** We consider in this section the transcription of a selection of real piano recordings, obtained from the MAPS dataset (Emiya et al., 2010). The data comes with a ground-truth binary "piano-roll" which indicates the active notes at every time. The note fundamental frequencies are given in MIDI, a standard musical integer-valued frequency scale that matches the keys of a piano, with 12 half-tones (i.e., piano keys) per octave. The spectrogram of each recording is computed with a Hann window of size 93-ms and $50\%$ overlap ($f_s$ = 44.1Hz). The columns (time frames) are then normalised to produce **V**. Each recording is decomposed with PLCA, OST and $OST_e$, with $K = 60$ notes (5 octaves). Half of the recording is used for validation of the hyper-parameters and the other half is used as test data. For PLCA, we validated 4 and 3 values of the width and amplitude dampening of the Gaussian kernels used to synthesise the dictionary. For OST, we set $\epsilon = q\epsilon_0$ in Eq. (4), which was found to satisfactorily improve the discrimination of octaves increasingly with frequency, and validated 5 orders of magnitude of $\epsilon_0$. For $OST_e$, we additionally validated 4 orders of magnitude of $\lambda_e$. Each of the three methods returns an estimate of **H**. The estimate is turned into a 0/1 piano-roll by only retaining the support of its $P_n$ maximum entries at every frame $n$, where $P_n$ is the ground-truth number of notes played in frame $n$. The estimated piano-roll is then numerically compared to its ground truth using the F-measure, a global recognition measure which accounts both for precision and recall and which is bounded between 0 (critically wrong) and 1 (perfect recognition). Our evaluation framework follows standard practice in music transcription evaluation, see for example (Daniel et al., 2008). As detailed in the supplementary material, it can be shown that $OST_g$ and $OST_{e+g}$ do not change the location of the maximum entries in the estimates of **H** returned by OST and $OST_e$, respectively, but only their amplitude. As such, they lead to the same F-measures than OST and $OST_e$, and we did not include them in the experiments of this section.

We first illustrate the complexity of real-data spectra in Fig. 4, where the amplitudes of the first six partials (the components corresponding to the harmonic frequencies) of a single piano note are represented along time. Depending on the partial order $q$, the amplitude evolves with asynchronous beats and with various slopes. This behaviour is characteristic of piano sounds in which each note comes from the vibration of up to three coupled strings. As a consequence, the spectral envelope of such notes cannot be well modelled by a fixed amplitude pattern. Fig. 4 shows that, thanks to its flexibility, $OST_e$ can perfectly recover the true fundamental frequency (MIDI 50) while PLCA

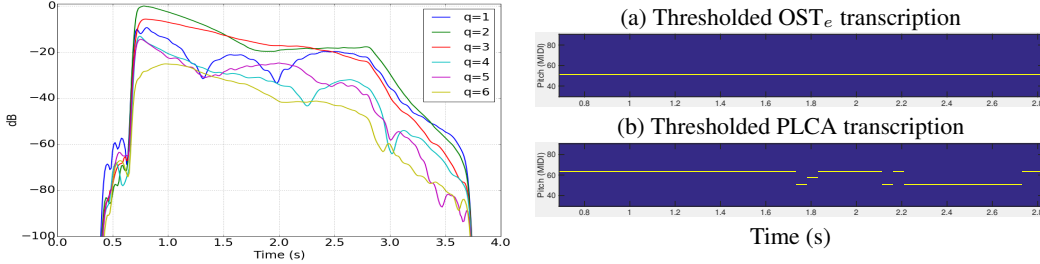

Figure 4: First 6 partials and transcription of a single piano note (note D3, $\nu_0$ = 147 Hz, MIDI 50).

Table 1: Recognition performance (F-measure values) and average computational unmixing times.

| MAPS dataset file IDs | PLCA | PLCA+noise | OST | OST+noise | $OST_e$ | $OST_e$+noise |
|---|---|---|---|---|---|---|
| *chpn_op25_e4_ENSTDkAm* | 0.679 | 0.671 | 0.566 | 0.564 | **0.695** | **0.695** |
| *mond_2_SptkBGAm* | 0.616 | **0.713** | 0.470 | 0.534 | 0.610 | 0.607 |
| *mond_2_SptkBGCl* | 0.645 | 0.687 | 0.583 | 0.676 | 0.695 | **0.730** |
| *muss_1_ENSTDkAm* 4 | 0.613 | 0.478 | 0.513 | 0.550 | **0.671** | 0.667 |
| *muss_2_AkPnCGdD* | 0.587 | 0.574 | 0.531 | 0.611 | 0.667 | **0.675** |
| *mz_311_1_ENSTDkCl* | 0.561 | 0.593 | 0.580 | 0.628 | 0.625 | **0.665** |
| *mz_311_1_StbgTGd2* | 0.663 | 0.617 | 0.701 | 0.718 | **0.747** | 0.747 |
| Average | 0.624 | 0.619 | 0.563 | 0.612 | 0.673 | **0.684** |
| Time (s) | 14.861 | 15.420 | **0.004** | **0.005** | 0.210 | 0.202 |

is prone to octave errors (confusions between MIDI 50 and MIDI 62). Then, Table 1 reports the F-measures returned by the three competing approaches on seven 15-s extracts of pieces from Chopin, Beethoven, Mussorgski and Mozart. For each of the three methods, we have also included a variant that incorporates a flat component in the dictionary that can account for noise or non-harmonic components. In PLCA, this merely consists in adding a constant vector $w_{f(K+1)} = 1/M$ to $\mathbf{W}$. In OST or $OST_e$ this consists in adding a constant column to $\widetilde{\mathbf{C}}$, whose amplitude has also been validated over 3 orders of magnitude. OST performs comparably or slightly inferiorly to PLCA but with an impressive gain in computational time ($\sim$3000$\times$ speedup). Best overall performance is obtained with $OST_e$+noise with an average $\sim$10% performance gain over PLCA and $\sim$750$\times$ speedup.

A Python implementation of OST and real-time demonstrator are available at `https://github.com/rflamary/OST`

## 7  Conclusions

In this paper we have introduced a new paradigm for spectral dictionary-based music transcription. As compared to state-of-the-art approaches, we have proposed a *holistic* measure of fit which is robust to local and harmonically-related displacements of frequency energies. It is based on a new form of transportation cost matrix that takes into account the inherent harmonic structure of musical signals. The proposed transportation cost matrix allows in turn to use a simplistic dictionary composed of Dirac vectors placed at the target fundamental frequencies, eliminating the problem of choosing a meaningful dictionary. Experimental results have shown the robustness and accuracy of the proposed approach, which strikingly does not come at the price of computational efficiency. Instead, the particular structure of the dictionary allows for a simple algorithm that is way faster than state-of-the-art NMF-like approaches. The proposed approach offers new foundations, with promising results and room for improvement. In particular, we believe exciting avenues of research concern the learning of $\mathbf{C}_h$ from examples and extensions to other areas such as in remote sensing, using application-specific forms of $\mathbf{C}$.

**Acknowledgments.** This work is supported in part by the European Research Council (ERC) under the European Union's Horizon 2020 research & innovation programme (project FACTORY) and by the French ANR JCJC program MAD (ANR-14-CE27-0002). Many thanks to Antony Schutz for generating & providing some of the musical data.

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
