[Supplementary Material]

# Optimal spectral transportation with application to music transcription – supplementary material

**Rémi Flamary**
Université Côte d'Azur, CNRS, OCA
`remi.flamary@unice.fr`

**Cédric Févotte**
CNRS & IRIT, Toulouse
`cedric.fevotte@irit.fr`

**Nicolas Courty**
University of Bretagne Sud, CNRS, IRISA
`courty@univ-ubs.fr`

**Valentin Emiya**
Aix-Marseille University, CNRS, LIF
`valentin.emiya@lif.univ-mrs.fr`

We provide in this supplementary material more details about some of the algorithmic aspects of OST.

**Optimisation for OST.** As discussed in the main paper, when no regularisation is used, OST unmixing boils down to the application of a linear operator $\mathbf{L}$ to the spectral samples, such that $\mathbf{h}_n = \mathbf{L}^\top \mathbf{v}_n$. $\mathbf{L}$ is a sparse matrix, with only $M$ nonzero coefficients among $KM$. It only depends on the transportation cost matrix $\widetilde{\mathbf{C}}$ and can be pre-computed like in Algorithm 1.

**OST with entropic regularisation (OST$_e$).** Entropic regularisation consists in penalising OST with the negentropic term $\Omega_e(\widetilde{\mathbf{T}}) = \sum_{ik} t_{ik} \log(t_{ik})$ weighted by $\lambda_e$. Remarkably, Benamou et al. (2015) have shown that this optimisation problem becomes:

$$\min_{\mathbf{h} \geq 0, \widetilde{\mathbf{T}} \geq 0} \quad D_{\mathrm{KL}}(\widetilde{\mathbf{T}}, \mathbf{Z}) \quad \text{s.t.} \quad \widetilde{\mathbf{T}}\mathbf{1}_K = \mathbf{v}, \quad \widetilde{\mathbf{T}}^\top \mathbf{1}_M = \mathbf{h}, \tag{1}$$

where $\mathbf{Z} = \exp(-\widetilde{\mathbf{C}}/\lambda_e)$, the exponential being being applied entry-wise. Again, because $\mathbf{h}$ is a variable of the optimisation problem, only $\widetilde{\mathbf{T}}$ needs to be optimised with $\mathbf{h}$ being subsequently computed by the second constraint in Eq. (1). The resulting problem consists of a Bregman projection and the optimal $\widetilde{\mathbf{T}}$ can be obtained by a simple scaling of the $\mathbf{Z}$ matrix (Benamou et al., 2015):

$$\widehat{\widetilde{\mathbf{T}}} = \mathrm{diag}\left( \frac{\mathbf{v}}{\mathbf{Z}\,\mathbf{1}_K} \right) \mathbf{Z} = \mathrm{diag}(\mathbf{v})\,\mathbf{L}_e, \tag{2}$$

where $\mathbf{L}_e$ is the matrix with coefficients

$$l_{ik} = \frac{\exp(-\tilde{c}_{ik}/\lambda_e)}{\sum_p \exp(-\tilde{c}_{ip}/\lambda_e)}. \tag{3}$$

Then, $\mathbf{h}$ is recovered from the second constraint in Eq. (1):

$$\mathbf{h} = \widehat{\widetilde{\mathbf{T}}}^\top \mathbf{1}_M = \mathbf{L}_e^\top \mathrm{diag}(\mathbf{v})\mathbf{1}_M = \mathbf{L}_e^\top \mathbf{v}. \tag{4}$$

**OST with group regularisation (OST$_g$).** The problem here considered is

$$\min_{\mathbf{h} \geq 0, \widetilde{\mathbf{T}} \geq 0} \quad \langle \widetilde{\mathbf{T}}, \widetilde{\mathbf{C}} \rangle + \lambda_g\,\Omega_g(\widetilde{\mathbf{T}}) \quad \text{s.t.} \quad \widetilde{\mathbf{T}}\mathbf{1}_K = \mathbf{v}, \quad \widetilde{\mathbf{T}}^\top \mathbf{1}_M = \mathbf{h}, \tag{5}$$

where

$$\Omega_g(\widetilde{\mathbf{T}}) = \sum_{k=1}^{K} \sqrt{\|\widetilde{\mathbf{t}}_k\|_1} \tag{6}$$

---

**Algorithm 1** Computation of OST's labelling matrix $\mathbf{L}$

---

**Require:** Transportation cost matrix $\widetilde{\mathbf{C}}$.
1: Initialise $\mathbf{L} = \mathbf{0}_{M \times K}$
2: **for** $i = 1, \ldots, M$ **do**
3:    $k_i^\star = \arg\min_k \{\tilde{c}_{ik}\}$
4:    $l_{ik_i^\star} = 1$
5: **end for**

---

**Algorithm 2** Unmixing with $\text{OST}_g$ (one sample)

---

**Require:** Sample $\mathbf{v}$, transportation cost matrix $\widetilde{\mathbf{C}}$, hyper-parameter $\lambda_g$.
1: Initialise $\widetilde{\mathbf{R}}^{(0)}$ with zeros, set $iter = 0$
2: **repeat**
3:    $iter = iter + 1$
4:    Compute $\widetilde{\mathbf{C}}^{(iter)} = \widetilde{\mathbf{C}} + \widetilde{\mathbf{R}}^{(iter)}$ with $\widetilde{\mathbf{R}}^{(iter)}$ computed with Eq. (8)
5:    Compute $\mathbf{L}_g^{(iter)}$ with Algorithm 1 applied to $\widetilde{\mathbf{C}}^{(iter)}$
6:    Compute $\widetilde{\mathbf{T}}^{(iter)} = \text{diag}\,(\mathbf{v})\mathbf{L}_g^{(iter)}$ and $\mathbf{h}^{(iter)} = \mathbf{L}_g^{(iter)\top}\mathbf{v}$
7: **until** convergence

---

and $\lambda_g$ is an hyper-parameter. As mentioned in the main paper, the optimisation problem may be addressed with majorisation-minimisation (MM). MM consists of iteratively minimising an upper bound of the objective function which is tight at the current iterate (Hunter and Lange, 2004). With this procedure, the objective function is guaranteed to decrease at every iteration. As it appears, in our case, we only need to majorise the penalty term $\Omega_g(\widetilde{\mathbf{T}})$ to obtain a tractable update. We may use the tangent inequality

$$\sqrt{\|\widetilde{\mathbf{t}}_k\|_1} \leq \sqrt{\|\widetilde{\mathbf{t}}_k^{(iter)}\|_1} + \frac{1}{2\sqrt{\|\widetilde{\mathbf{t}}_k^{(iter)}\|_1}}(\widetilde{\mathbf{t}}_k - \widetilde{\mathbf{t}}_k^{(iter)})^\top \mathbf{1}_M \tag{7}$$

where $\widetilde{\mathbf{t}}_k^{(iter)}$ is the current estimate. The inequality essentially linearises the regularisation term, whose contribution can now be absorbed into the inner product $\langle \widetilde{\mathbf{T}}, \widetilde{\mathbf{C}} \rangle$, by replacing $\widetilde{\mathbf{C}}$ with $\widetilde{\mathbf{C}}^{(iter)} = \widetilde{\mathbf{C}} + \widetilde{\mathbf{R}}^{(iter)}$ where $\widetilde{\mathbf{R}}^{(iter)}$ is the $M \times K$ matrix with coefficients

$$\widetilde{r}_{ik}^{(iter)} = \frac{1}{2}\|\widetilde{\mathbf{t}}_k^{(iter)}\|_1^{-\frac{1}{2}}. \tag{8}$$

The resulting overall optimisation procedure is summarised in Algorithm 2.

**Group-regularisation does not change the maxima locations of h.** $\text{OST}_g$ is a nonconvex problem (because $\Omega_g(\widetilde{\mathbf{T}})$ is a nonconvex function). Because MM is only a descent procedure, the solution will be dependent on the initialisation. By initialising $\widetilde{\mathbf{R}}^{(0)}$ with zeros in Algorithm 2 we ensure that the first estimate $\widetilde{\mathbf{T}}^{(1)}$ is the solution of the unregularised OST. The expression of $\widetilde{\mathbf{R}}^{(iter)}$ given by Eq. (8) shows that columns of $\widetilde{\mathbf{T}}$ with larger norms will subsequently be less penalised than those with smaller norms. This promotes group-sparsity along iterations but has another interesting property. Indeed it is easy to see that the ranking of the values of the norms will not change along iterations. Even though the group regularisation promotes sparsity and affects the magnitude of the coefficients in $\mathbf{h}$, one can predict from initialisation in which order the columns may shrink to zeros. As such the locations of the maxima will be same in the estimates of $\mathbf{h}$ returned by OST or $\text{OST}_g$. Hence, the F-measures will be same for the two methods in the experimental results with real data, and only OST needs to be considered, *as far as this particular evaluation metric is considered.*

**Doubly-regularised OST ($\text{OST}_{e+g}$).** Doubly-regularised OST may be addressed in the same MM setting, again by linearising $\Omega_g(\widetilde{\mathbf{T}})$. Then, Algorithm 2 applies again by computing $\mathbf{L}_g^{(iter)}$ using Eq. (3) instead of Algorithm 1. For this same reason, the location of the maxima in $\mathbf{h}$ as returned by $\text{OST}_{e+g}$ will also be the same as those returned by $\text{OST}_e$.