[Reviews · NeurIPS 2016]

Reviewer 1

Summary

This paper concerns music transcription basic on a matrix decomposition approach. The novelty of the paper is to define a divergence measure based on optimal transportation, which allows shifts of energy to neighbouring frequencies. In addition the measure also allows energy to be transferred at no cost between spectral harmonics. Experimental results are presented on a set of seven 15s extracts of piano music.

Qualitative Assessment

I enjoyed reading this paper. It is well-motivated and the optimal transportation measure is a very nice approach which puts the modelling of variation into the divergence rather than the learned dictionary. The approach is well-explained. The experiments seem very small-scale (less than 2 minutes of music in total) and although comparisons are given against PLCA, I'm not convinced that this gives a meaningful comparison with the state-of-the-art. So I would say this a proof-of-concept, rather than work that permits strong conclusions. The paper is well-structured, but there are quite a few grammatical errors, so it could benefit from careful proof-reading, and some re-phrasing. Also write "timbre" rather than "timber".

Confidence in this Review

1-Less confident (might not have understood significant parts)


Reviewer 2

Summary

This paper proposes a new spectral unmixing system for music transcription which has the following highlights: 1. The authors replace the KL divergence in PLCA with OT divergence, which can embed a form of invariance to displacements of support, as defined by the transportation cost matrix C. 2. The authors proposes a harmonic-invariant transportation cost C, which allows a mass at one frequency to be transported to a divisor frequency with no cost. 3. Because of 1 and 2 above, the authors are able to use Dirac vectors directly as the spectral dictionary, which saves the huge labour of designing a meaningful dictionary. The authors also provide optimization method for the proposed system, and evaluate it on both toy data and real musical data, and prove the effectiveness of the proposed system.

Qualitative Assessment

1. In line 63 and 73, the authors explain why KL divergence in PLCA is not optimal in the case of music transcription. In my opinion, it might be easier if the authors use some figures or numbers to "quantify" the explanation. 2. In formula (4), the authors use a quadratic cost for different frequencies. Could you add more explanation of why quadratic is used? Why not absolute value or some other cost? 3. In Section 6.2, the authors briefly mention that they also use a flat component in the dictionary that can account for noise or non-harmonic components, which seems to give better performance in the experiments. Please also explain why this flat component should help in the presence of noise.

Confidence in this Review

2-Confident (read it all; understood it all reasonably well)


Reviewer 3

Summary

The paper proposes a spectral unmixing method for automatic music transcription, realted to NMF/PLCA methods but replacing the standard divergence methods by Optimal Transport (OT), similar to the use of an "earth mover distance". The transportation cost is constructed to have a zero cost to map mass at harmonic frequencies to the fundamental.

Qualitative Assessment

The paper is interesting, well written, and described in an appropriate context of state of the arts for NMF. Transportation distances are not unknown in music, see e.g. [1] (used for symbolic matching). [1] Typke, R., Veltkamp, R. C., & Wiering, F. (2004, October). Searching notated polyphonic music using transportation distances. In Proceedings of the 12th annual ACM international conference on Multimedia (pp. 128-135). ACM. The algorithm has comparable performance to the traditional PLCA approach (better in most cases), with a substantial performance improvement of 2 or more orders of magnitude. The authors may wish to contrast their approach to the use of Chroma features, which is well known in music information retrieval: the frequency wrapping may have a similar effect.

Confidence in this Review

2-Confident (read it all; understood it all reasonably well)


Reviewer 4

Summary

In this paper, spectral unmixing problem for music transcription is formulated as a matrix factorization problem like NMF. In contrast to existing approaches, e.g., PLCA that minimizes KL-divergence between an original spectrogram and multiplication of decomposed factors, authors take into account a small displacement in frequency domain and modeled this displacement by using a transportation matrix. The spectral unmixing is formulated as an optimization problem that minimizes a total transportation cost for a given dictionary matrix. This can be regarded as a natural extension of EMD-NMF which is special case of the proposed method when a cost matrix C is a symmetric matrix. The proposed cost function can be efficiently minimized by a closed-form solution if a set of Dirac vectors are used for a dictionary matrix. In the experiments, authors demonstrated that OST with regularization outperformed PLCA in terms of both accuracy and efficiency.

Qualitative Assessment

-Technical correctness and novelty The ideas of modeling frequency displacement by a transportation matrix and obtaining the optimal displacement by closed-form solution are very interesting. (But I am not sure if these ideas have already been proposed in preceding studies or not.) It is also interesting that a harmonic nature can be naturally modeled by a cost matrix C. As suggested by authors, much further investigation on training C from data is an interesting direction as a future work. From a critical point of view, authors do not show a certain cases where group regularization is beneficial for real data. As long as seeing section 6.2 and Table 1, OST with entropic regularization is sufficient to deal with real data. Much deeper investigation on group regularization for real-data is desired. From another aspect, I am a little concerned about the fact that the dictionary matrix is completely hand-crafted and the frequency displacement model is also data-independent. Can OST outperform methods that use data-dependent dictionary that trained from labeled data? -Usefulness It is very interesting that there exists closed form solution for OST. Thanks to this fact, the proposed method achieves significantly faster processing compared to PLCA. Reviewer thinks that this characteristic is very practical for real applications. On the other hand, I am not sure if OST really outperforms any other preceding works in terms of accuracy, because OST was compared with only a single method, PLCA, which was proposed in 2006 and may be outdated. -Clarity This paper is quite well written. Although I do not confident in acoustic signal processing (instead, I confident in matrix factorization problem including NMF), it was very easy to understand technical details. The proposed procedure is correctly formulated by mathematical notations. I found a minor mistake in line 3-4 in Section 3. The variable 'N' in the following sentence, i,j=1, ..., N, must be 'M'. %% After Rebuttal %% The reviewer thanks to the authors for their detailed comments. I agree that this paper places importance on the theory part rather than experiments for the practical usefulness. This is a very good paper which should be published from NIPS.

Confidence in this Review

2-Confident (read it all; understood it all reasonably well)


Reviewer 5

Summary

This paper studies the problem of transcribing music using non-negative factorizations of spectrograms. The authors propose using what is effectively a transportation cost as a measure of fit, rather than the state-of-the-art that uses KL divergence. They design the transportation cost matrix to take into account a number of issues specific to real-life music, including deviations from fundamental frequencies, variation in amplitude at harmonics, and sampling rates. Experiments on synthetic and real music show that certain variations of their approach outperform the state-of-the-art in both speed and accuracy.

Qualitative Assessment

The approach is very nice, and logically seems more well-suited to music transcription than the PLCA method. Most of the design of the algorithm comes from general knowledge of music though, as opposed to hard data, and there is no theoretical evidence provided for the performance of this approach. A few comments: 1. The paper builds very slowly; the OST algorithm is not fully described until page 5, and evidence for its performance is not provided until the experiments on page 7. It would be useful to state and summarize the overall contributions early on, ideally in the first section. 2. The entire algorithm should be written in pseudocode at some point, not just described in the text. 3. The experiments establish that OST and its regularized variants are all much faster than PLCA. In terms of performance though, only the regularized versions seem to consistently outperform PLCA in accuracy. The regularization should be presented as a key part of the main algorithm, not just as variants.

Confidence in this Review

1-Less confident (might not have understood significant parts)


Reviewer 6

Summary

REPLACE THIS WITH YOUR ANSWER

Qualitative Assessment

Major concerns are as follows: 1) For a given W, the problem (1) or (3) are both convex and easy enough to deal with. The significance of the proposed approach is not clear. Also, since the fitting criterion in (2) and (3) were both proposed in some other papers, the contribution of the current submission seems merely applying this technique to the music application, which is not significant. 2) As mentioned in the paper, there are many algorithms that deal with the robustness issue of this particular problem, for example, Vincent et al., 2010 and Rigaud, et al., 2013. Comparison between these methods are missing. 3) The title “optimal spectral transportation” is rather unclear: it is hard to see where does the optimality lie in and it was not explained in the paper. 4) The usage of a Dirac W is confusing. In fact, using a Dirac W means that the goal reduces to matching Vtilde and H under the dictionary Wtilde = identity, where _tilde corresponds to the K nonzero frequencies. But for a valid fitting criterion, the optimal solution of the above seems to be simply H = Vtilde. The authors did not explain how to avoid this trivial solution.

Confidence in this Review

2-Confident (read it all; understood it all reasonably well)